materials science/synthetic chemistry/spectroscopy

palm olein, recycled PET, polyols, PU coatings, anticorrosion

**Author for correspondence:**
Seng Neon Gan
e-mail: sngan@um.edu.my

This article has been edited by the Royal Society of Chemistry, including the commissioning, peer review process and editorial aspects up to the point of acceptance.

# Thermal and anticorrosion properties of polyurethane coatings derived from recycled polyethylene terephthalate and palm olein-based polyols

## Abbas Ahmad Adamu[1,2], Norazilawati Muhamad Sarih[1] and Seng Neon Gan[1]

[1]Department of Chemistry, University of Malaya, 50603 Kuala Lumpur, Malaysia
[2]Department of Polymer Technology, Hussaini Adamu Federal Polytechnic, 5004 Kazaure, Nigeria

AAA, 0000-0003-3519-9251; NMS, 0000-0001-7850-1049; SNG, 0000-0002-6686-2602

Polyols of palm olein/polyethylene terephthalate (PET) were synthesized by means of incorporating recycled PET from waste drinking bottles in different proportions into palm olein alkyd in the presence of ethylene glycol. The polyols were characterized by FTIR, and theirs hydroxyl value (OHV), acid value (AV) and viscosity were determined. The formulation of the polyurethane coating was carried out by dissolving the polyol in mixed solvent of cyclohexanone/tetrahydrofuran (THF) (4 : 1) followed by reacting 1 hydroxyl equivalent of the polyol with 1.2 equivalents of methylene diphenyldiisocyanate and 0.05% dibutyltin dilaurate (DBTDL) catalyst. The coating cured through the cross-linking reactions between hydroxyl and isocyanate groups. The formation of urethane linkages was established by FTIR spectroscopy. The set films were characterized by thermal analysis. To study their anticorrosion properties, polarization measurements and EIS in 3.5% NaCl solution were determined. The coatings displayed good thermal stability and anticorrosion properties which were supported by XRD analysis. The PU7 coating, with the highest proportion of PET (up to 15% w/w), displayed significantly improved thermal stability and anticorrosion properties. It is evident that the performance of the polyurethane (PU) coatings could be enhanced by the incorporation of PET.

# 1. Introduction

The increasing production and usage of recycled polyethylene terephthalate (PET) in soft drink bottles have generated grievous environmental problems owing to its non-biodegradable nature [1]. The relatively large molecules of PET cannot be easily broken down by known microorganisms, therefore, complex and expensive techniques are needed to render them biologically degradable. Recycling of PET bottles is necessary not only for environmental protection issues, but also for the recovery of valuable products [2]. Many ways of managing the PET waste have been developed, such as (i) reduction of the waste generation, (ii) incineration, (iii) bio or photodegradation, (iv) composting and (v) recycling. Among these, recycling is the favourite solution [1]. Among the different recycling techniques, chemical recycling would be more attractive, as it would help in the recuperation of the raw materials used in producing the polymer, and other valuable secondary products [3,4]. PET recycling is not only used as a remedy to decrease the problem associated with solid waste, it also serves in the acquiring of petrochemical products and energy.

PET bottles are now one of the most valued and successful recyclable materials [3]. The recycling of PET by the chemical method can be conducted by processes such as glycolysis, hydrolysis, alcoholysis, aminolysis and simultaneous hydrolysis [5].

Vegetable oils (VO) and their derived fatty acids are environmentally friendly raw materials, and they are widely used in the synthesis of alkyd resins. Oils improve the flexibility and solubility of the coatings and reduce the brittleness of the film [6]. The advent of VO for preparation of alkyd resins and polyurethanes is owing to their availability, moderately less cost, exclusive chemical structure, functionality, reactivity, low toxicity and for being biodegradable [7,8]. The exceptional characteristics of VO are their exceptional chemical arrangements such as unsaturation sites, epoxies, hydroxyls and esters alongside intrinsic fluidity features. They could be subjected to different chemical changes to produce new polymers that have vast uses, particularly for paints and coatings [9].

Alkyds are the main synthetic resins extensively used in the paint industry because of their vast and worthy output [10,11]. Alkyd-based coatings are recognized for their excellent performance ranging from good corrosion protection, high gloss and easiness of application irrespective of the nature of the applied surfaces [12,13].

Polyurethanes are polymers having urethane linkages. They are prepared by step-growth polymerization of isocyanate groups with hydroxyl groups. Polyurethanes are of various types and have a diversity of applications ranging from coatings, adhesives, shoe soles, foams for mattresses and insulation [14]. They are mostly produced from polyether and polyester polyols derived from petrochemicals [15]. Due to their excellent adhesion to diverse types of surfaces, abrasion resistance and electrical insulation and durability, chemical and corrosion resistance, polyurethanes are also widely used as coatings [16]. Corrosion is considered as a gradual exhaustion of metals through the chemical or oxidizing process [17]. Various methods for protection against corrosion were developed and studied [18]. However, the coating systems could be hazardous to our environment if they comprise harmful solvent and toxic components. Hence, new anticorrosion coatings that are more environment friendly and less toxic would be desirous.

Karayannidis et al. [19] reported the recycling of PET waste bottles by depolymerization using DEG to acquire glycolysates which were further reacted with maleic anhydride, phthalic anhydride (PA) and PG to prepare unsaturated polyesters that are subsequently cured by mixing with styrene and benzoyl peroxide/amine initiator system. Atta et al. [3] have reported the anticorrosion coating prepared from PET waste for carbon steel. The coatings were assessed for corrosion resistance based on salt spray and cathodic disbondment. Ahmad et al. [17] have reported the synthesis of eco-friendly alkyd nanocomposite coatings, and the influence of nanofillers on the performance properties of the coatings were evaluated. The coatings displayed excellent anticorrosion properties. Ang et al. [20] have reported polyester from reacting phthalic acid with polyol synthesized from ring-opening reaction of the epoxidized palm olein. For the preparation of polyurethane adhesive, the polyol was reacted with polymeric methylene diphenyl diisocyanate (MDI) at isocyanate index of 1.3. Singh et al. [21] have prepared waterborne anticorrosive coating from hyperbranched polyester polyol. Narayan et al. [22] had prepared dispersions of cross-linked polyurethane for anticorrosion coatings. The cross-linked films were evaluated for the influence of different diisocyanate and acetoacetylation on the stability of reactive dispersion and properties. Marathe et al. [23] studied the synthesis of neem acetylated polyester polyol as environment friendly polyurethane (PU) coatings. The incorporation of the quinoline encapsulated corrosion inhibitor into the PU coatings shows improved performances. Velayutham et al. [24] reported the synthesis of glycerol, PA and oleic acid polyols-based PU coatings. The coatings were characterized, and the formulation can be carefully controlled to obtain the desired performance. Anand et al. [25] reported that synthesized sorbitol-based polyols to prepare ZnO-reinforced PU coatings. Balgude et al.

[26] studied the preparation of aqueous 2 K-polyurethane coatings from the cardanol derivatives to form water-based polyols as a potential renewable resource. The chemical resistance, thermal stability and corrosion resistance of the polyurethane coatings have been greatly influenced by cross-link densities.

In the present work, an alkyd was synthesized from palm olein, glycerol and PA. Waste PET from soft drink bottles and ethylene glycol (EG) were introduced into the palm olein alkyd. Simultaneous depolymerisation of PET by EG and transesterification with alkyd occur at high temperature to form a viscous product (palm olein/PET polyols). The new polyol was formulated to produce an anticorrosion polyurethane coating. We are the first to report the direct incorporation of PET into a palm oil-based alkyd to form polyol for making new polyurethane coatings.

# 2. Experimental

## 2.1. Materials

The waste PET used in the synthesis was from soft drink bottles that were washed and cut into small pieces of about $2 \times 2$ mm. Palm olein (99.5%), glycerine (99.5%) and EG were obtained from Emery Oleochemicals Sdn Bhd, Malaysia, PA from DC Chemicals Korea, sodium hydroxide (NaOH) from R & M Marketing, Essex, UK, cyclohexanone and tetrahydrofuran (THF) from Merck KGaA, Marmstard, Germany. MDI and dibutyltin dilaurate (DBTDL) were from BASF Chemicals and used as received.

## 2.2. Methods

### 2.2.1. Synthesis of palm olein/PET polyols

The reaction was carried out in a 1 l reactor flask furnished with a thermometer, a reflux condenser, Dean–Stark distilling trap, and a mechanical stirrer. A 474 g palm olein, 200 g glycerol and 1 g NaOH were introduced into the reactor. These were stirred and heated at 220°C for 3 h to produce predominantly a mixture of monoglycerides. The reactor was cooled down to 120°C and 135 g PA was added and the mixture heated and held at 130°C for 1 h, the temperature was then slowly raised to 180°C, and then to 200°C for 1 h. Water evolved from the condensation reactions was removed by the Dean–Stark distilling trap. Forty-eight grams of EG and the specified amount of PET were subsequently introduced into the reactor and heated to 220–240°C. Simultaneous depolymerization of PET by EG and transesterification reactions with the alkyd occur to produce a viscous product (palm olein/PET alkyd polyols). This procedure was repeated for four different amounts of PET at 85.6, 106, 128, 132 g to produce four different palm olein/PET alkyd polyols designed as Alk 4, Alk 5, Alk 6, Alk 7, respectively. For control, a sample was prepared without the addition of PET and EG and was referred to as Alk C. Incorporation of PET into the alkyd has led to increase in the viscosity of the final polyol, and the polyol would become non-flowable if more than 132 g PET was used. The plausible reactions involved are represented in a schematic diagram (scheme 1).

## 2.3. Characterizations

### 2.3.1. FTIR spectroscopy

The polyols were analysed by FTIR using Perkin Elmer spectrum RX 1 KBR spectrometer from 4000 to 400 cm$^{-1}$.

### 2.3.2. Determination of hydroxyl value

The hydroxyl value (OHV) was determined in accordance to ASTM D4274, method B. According to this test method, the OHV is expressed as the number of milligrams of potassium hydroxide equal to the hydroxyl amount of 1 g of the material. This test method involves two stages. The first stage is the preparation of test reagents and standardization of the potassium hydroxide solution, while the next stage is titration of the sample and blank after treating with known amount of PA dissolved in pyridine.

### 2.3.3. Determination of acid value

The acid value of the polyols was obtained in accordance with the standard test procedure as described by ASTM D 97400. The acid value is described as the number of milligrams of KOH needed to neutralize the free acidic groups in 1 g of the sample. To achieve this test, two stages are required. The first stage is

**Scheme 1.** Plausible structure of palm olein/PET polyol.

the preparation of test reagents and standardization of ethanolic potassium hydroxide. The second stage is titration of the sample and blank.

### 2.3.4. Viscosity measurement

Measurement of viscosity against shear rate for the polyols were done in an Anton Paar MCR 301 rheometer at shear rate from 0.001 to 5000 $s^{-1}$ at 25°C.

## 2.4. Preparation and curing of PU coatings

PU coatings were prepared as follows. The polyol solutions were prepared in mixed solvent of cyclohexanone and THF (4 : 1). One hydroxyl equivalent of the polyol was mixed with 1.2 isocyanate equivalent of MDI and a tin catalyst (DBTDL, 0.05 wt%) to form the coating solution, which should be used within 1 h. Typically, 11 g of the polyol is dissolved in 27 g solvent, and mixed with 16 g MDI and 0.021 g tin catalyst to form a total of 59 g of the coating solution. Mild steel panels were smoothed with fine sand paper and cleaned with ethanol to remove any grease just before the coating solution was applied using a bar coater and cured in a fully ventilated fume cupboard for 5 days at room temperature. To ensure complete curing, the panels were left overnight in an oven at 50°C. The thickness of the coatings was 20 ± 1 μm measured with a Elcometer 456 coating thickness tester.

### 2.4.1. Thermogravimetric analysis

The PU film was analysed for thermal stability in nitrogen atmosphere using a Perkin Elmer TGA 4000 instrument. Sample of about 5–10 mg was placed in alumina crucible and heated from 30–900°C at 10°C $min^{-1}$. The coatings' stability and degradation were analysed.

**Table 1.** Properties of the alkyd polyols.

| polyol | % PET | OHV (mg KOH g$^{-1}$) | acid value (mg KOH g$^{-1}$) | viscosity (cPs) |
|---|---|---|---|---|
| Alk C | 0.0 | 449 | 14.9 | 1840 |
| Akh 4 | 9.0 | 521 | 15.5 | 3820 |
| Alk 5 | 11.0 | 530 | 16.0 | 4360 |
| Alk 6 | 13.0 | 536 | 16.0 | 5940 |
| Alk 7 | 15.0 | 547 | 16.3 | 11 600 |

### 2.4.2. Corrosion studies

During the corrosion test, a glass tube with 2 cm diameter and 4 cm long was fixed onto the coated panel and an electrolyte was added. Electrochemical impedance spectroscopy (EIS) is a prevailing technique used to evaluate the anticorrosion performance of coatings. EIS is most commonly run in three electrodes mode. The coated steel as working electrode (WE), a graphite rod and Ag/AgCl (SCE) as counter and reference electrodes, respectively.

### 2.4.3. Electrochemical method

The extent of impedance was applied to evaluate the behaviour of the coatings towards the passage of electrons and charges into the coated steel surface. The EIS experiment was used to assess the effects of different percentages of PET in various PU-coated mild steel panels. The impedance spectra were studied at frequency range of 100 kHz and 0.01 Hz. FRA.EXE software connected to USB_IF030 interface in a computer was used in the Autolab PGSTAT302N, during the analysis in the EIS studies. To protect against electromagnetic interference, an electrochemical cell was located in a Faraday cage. Using a glass tube attached to each coated mild steel, a surface area of 3.14 cm$^2$ was exposed to solution of 3.5% NaCl. The study was conducted at room temperature by installing the analysis results in corresponding circuits via the nonlinear least-square fitting process. The fitting measured limiting $C^2$ value and by regulating comparative error worth of every component in the equal circuit to 5%. The Tafel measurement was conducted for using 3.5% NaCl solution as the electrolyte in the range of $-1.5$ to $+1.0$ V at 5 mV s$^{-1}$.

### 2.4.4. XRD analysis

X-ray diffraction study was carried out for PU-coated mild steel panels with Empyrean pan analytical diffractometer, using Cu-K$\alpha$ radiation, provided with a copper X-ray source. Copper bases emit X-rays with a wavelength, $\lambda = 1.5406$ Å.

Two samples were analysed by XRD to confirm that corrosion has taken place on the PU-coated panels. The first is an unexposed panel, and the second is one that has been exposed to 3.5% NaCl solution for 30 days.

# 3. Results and discussion

## 3.1. Hydroxyl value

The hydroxyl values of the alkyd polyol resins are presented in table 1. Alk C has the lowest OHV, while the others have higher OHV (around $535 \pm 10$) due to the additional EG introduced together with the PET. The % PET is calculated from the weight of PET used over the final weight of the polyols produced.

## 3.2. Acid value

The acid values of the polyols are summarized in table 1. Residual –COOH groups in the alkyd could react with amino group of polyurethane chain to produce amide, which could contribute to film

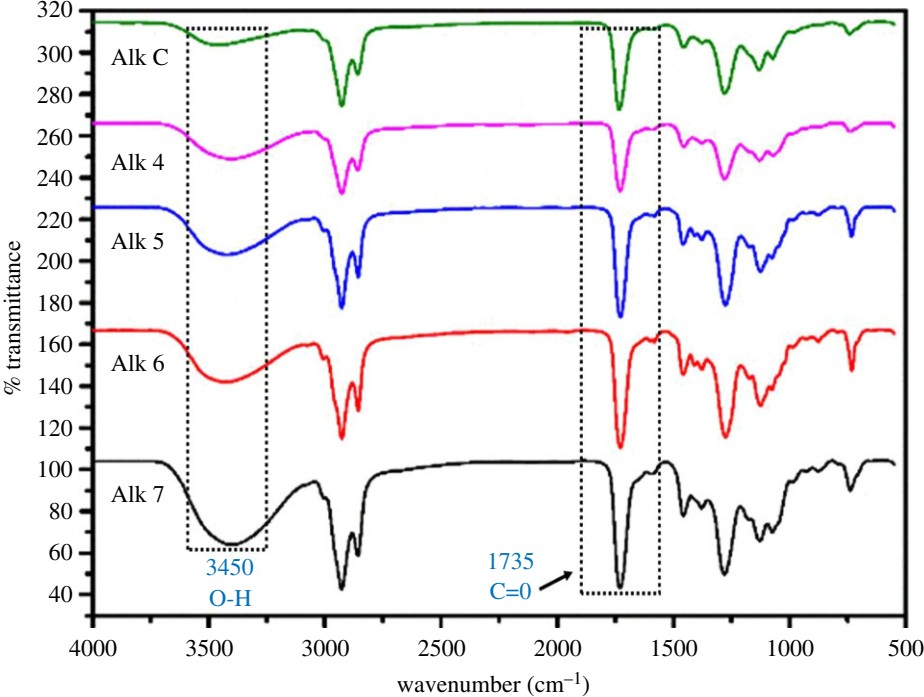

**Figure 1.** FTIR spectra of polyols.

hardness. However, within experimental errors, these alkyd polyols have acid value (AV) of $15.5 \pm 0.5$ mg KOH g$^{-1}$, which is less than 3% OHV, thus the effect to coating hardness would not be significant.

## 3.3. Viscosity

The rheological properties of polyols are among their most essential features. The functionality of polyols, such as the extent of branching, molar mass and molar mass distribution could influence the rheological properties of polymers [27]. Measurement of viscosities of the polyols were conducted in an Anton Paar MCR 301 rheometer at shear rate from 0.001 to 5000 s$^{-1}$ at 25°C. The viscosities of polyols are shown in table 1. The viscosity tends to increase as the functionality and molar mass increase. The alkyd polyol with the higher amount of PET has higher viscosity.

## 3.4. FTIR spectroscopy of polyols

FTIR spectra of the polyols are shown in figure 1. The –OH broad band is seen at 3450 cm$^{-1}$. A band close to 1735 cm$^{-1}$ is accredited to C=O group of ester. Bands near 1656 cm$^{-1}$ correspond to the aromatic ring. The –CH group bending vibrations present at frequencies near 1376 and 1458 cm$^{-1}$. The presence of a strong band at 1072 cm$^{-1}$ indicates that PET was incorporated into the polyol. However, the key bands of hydroxyl and carbonyl stretching are seen as broad peaks at 3450 and 1735 cm$^{-1}$, respectively, confirming the formation of polyester polyol [15,21].

## 3.5. Coatings formulations

The polyol solutions were prepared in mixed solvent of cyclohexanone and THF (4 : 1). One hydroxyl equivalent the polyol was reacted with 1.2 isocyanate equivalent amount of MDI in the presence of tin catalyst DBTDL (0.05% w/w). The coatings cured predominantly through the reactions between hydroxyl and isocyanate groups. The palm olein moieties contain unsaturated oleic acid that could also contribute to the cross-linking reaction by free radical mechanism [28] to influence the film dryness. All the coatings were fully cured, and then subjected to characterizations for thermal and anticorrosive properties.

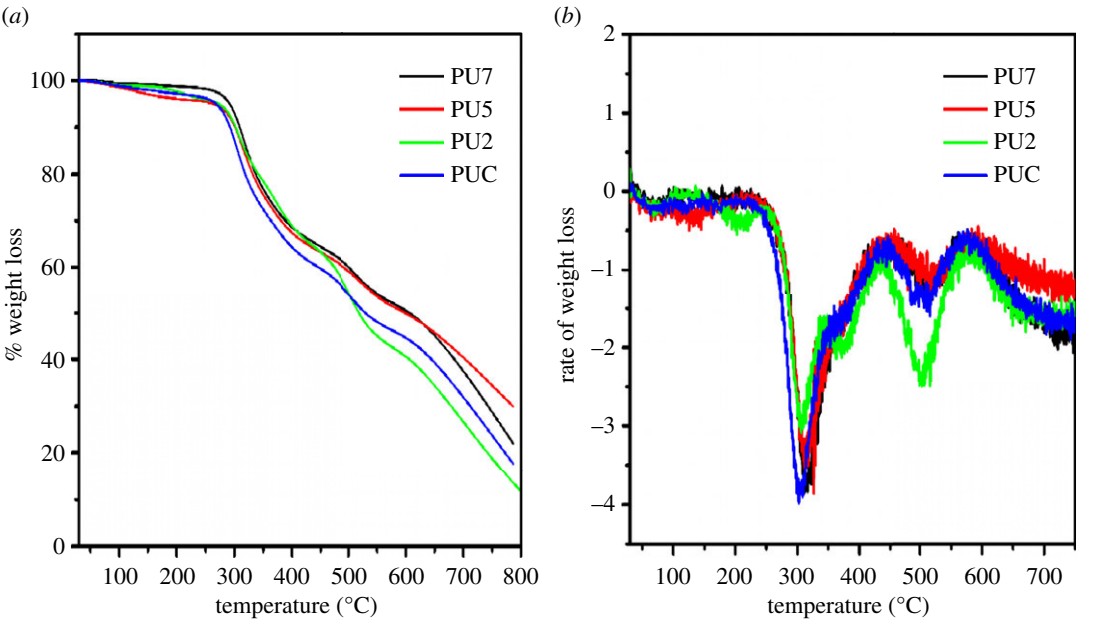

**Figure 2.** TGA/DTG curves of various PU-coated mild steel panels.

**Table 2.** Percentages of weight loss of PU coatings.

| sample | PUC | PU4 | PU5 | PU7 |
|---|---|---|---|---|
| $T_{5\%}$ (°C) | 264 | 267 | 272 | 291 |
| $T_{10\%}$ (°C) | 292 | 301 | 301 | 309 |
| $T_{20\%}$ (°C) | 318 | 331 | 337 | 341 |
| $T_{50\%}$ (°C) | 519 | 531 | 601 | 607 |
| % residue at 850°C | 18 | 23 | 28 | 34 |

## 3.6. Thermal properties of the PU coatings

Thermogravimetric analysis (TGA) was employed to assess the thermal stability of the coatings. The curve of % weight loss against temperature is referred to as the thermogravimetry TG curve and the rate of weight loss versus temperature is denoted as derivative thermogravimetric curve (DTG). The measurements were conducted in the nitrogen atmosphere at a scan rate of $10°C\ min^{-1}$ and the results are shown in figure 2. The % weight loss, temperatures and residues at the end of the thermal degradation at 750°C are presented in table 2. With reference to the DTG curves in figure 2, the first degradation step is mostly due to dissociation of the hard urethane segments in the PU coatings [23], whereas thermal degradations due to the scission of ester moieties and linear hydrocarbon chains occur in the second step [25]. The observed degradations in the third step were due to the C=C double bond cleavage. The samples were quite free of solvent or moisture as there was negligible weight loss at temperature below 150°C, and the breakdown of urethane bonds occurs above 250°C which leads to the formation of isocyanate, alcohol, $CO_2$, primary and secondary amines and olefins [17]. PU6 is very similar to PU5, presumably due to the fact that they have the same OHV and AV (table 1). The curves of PU6 overlap closely with those of PU5 and were excluded in figure 2. With reference to the results in table 2, the thermal stability of the coatings is in the order: PU7 > PU5 > PU4 > PUC. PU7 has exhibited higher thermal stability owing to higher degree of cross-linking [25].

## 3.7. Corrosion studies

### 3.7.1. Electrochemical impedance spectroscopy analysis

Figure 3 displays the Nyquist plots of the steel panels coated with PUC, PU4, PU5, PU6 and PU7 after 1, 15 and 30 days of immersion. Several orders of impedance were witnessed after 15 and 30 days of

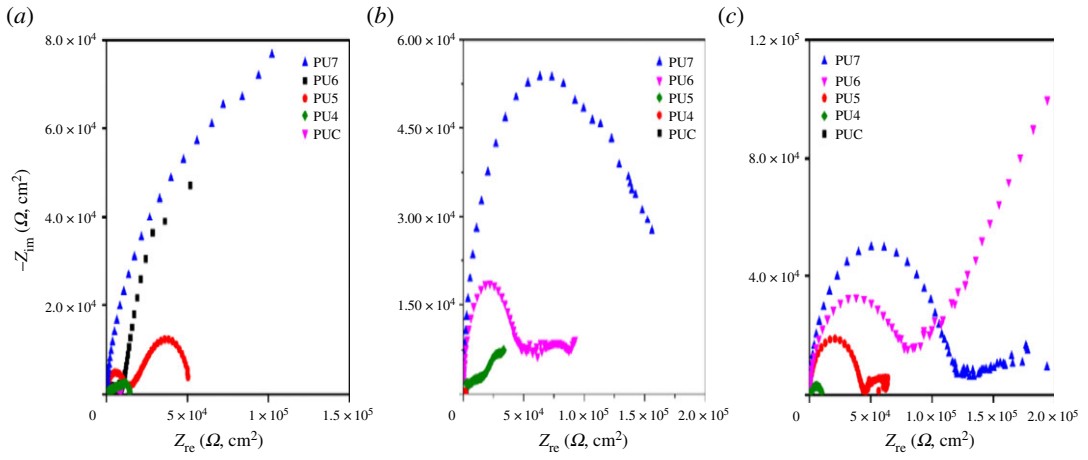

**Figure 3.** Nyquist plots of various PU-coated mild steel after soaking for (*a*) 1 day, (*b*) 15 days and (*c*) 30 days in a 3.5% NaCl solution.

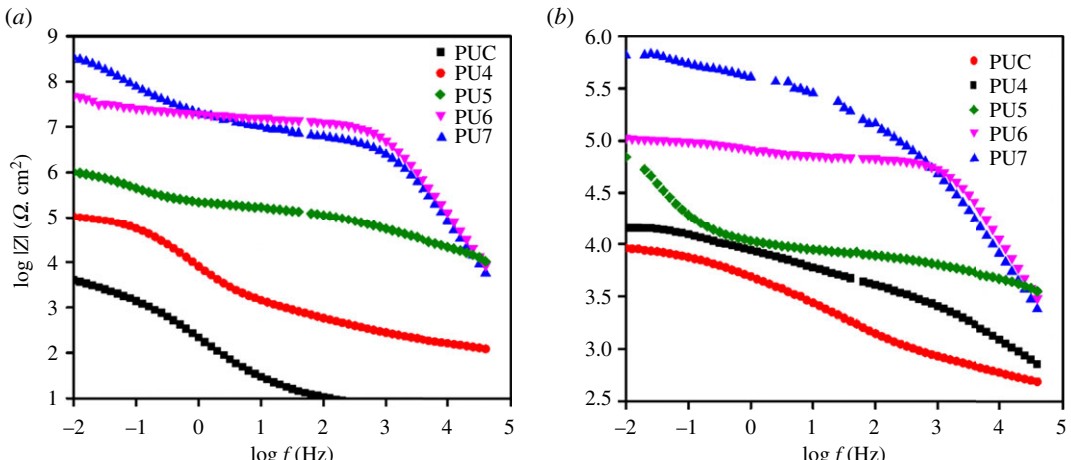

**Figure 4.** Bode plots of various PU coatings after soaking for (*a*) 15 days and (*b*) 30 days.

exposure. Most of the coatings after 15 days of exposure show two times constant. The smaller semicircle is linked to the process underneath the steel surface, while the bigger semicircle is related with the coating protection. PU7 coating displays only one semicircle for the same time of immersion (*b*). After 30 days of immersion (*c*), similar orders of magnitude in impedance as in after 15 days were observed, with most of the coated panels showing two times constant. This result revealed that the increase of the percentage of PET in the PU influences the increase in the real impedance ($Z_{re}$), with PU7 having higher impedance which is attributed to greater corrosion resistance [29]. The presence of PET in the PU coatings could enhance the properties of the coatings for surface corrosion protection and control.

Figure 4 shows the Bode modulus of various PU-coated mild steel panels, after 15 and 30 days of exposure. There was no change in the log [Z] value from day 1 to day 15 for all the coated steel samples. Figure 4*b* shows that with long soaking time, presumably the reduction in the value of log | Z| is owing to the diffusion of electrolyte into the steel [30]. Consequently, a substantial increase in coating capacitance and reduction in resistance of the material are noted.

Subsequently, with prolonged immersion, the log |Z| decreases from around 8.8 at the beginning of the immersion to about 5.8 after 30 days of immersion for PU7-coated steel, hence the rate of electrolyte penetration is gradual. PU7 coating, having the highest amount of incorporated PET, exhibits greater corrosion protection followed by PU6, which are better than the other coatings although all the other coatings show reasonable protection except for PUC, showing the poorest protection. This is owing to the barrier effects provided by the coatings preventing the penetration of oxygen and moisture into the surface of the steel leading to the development of a submissive layer.

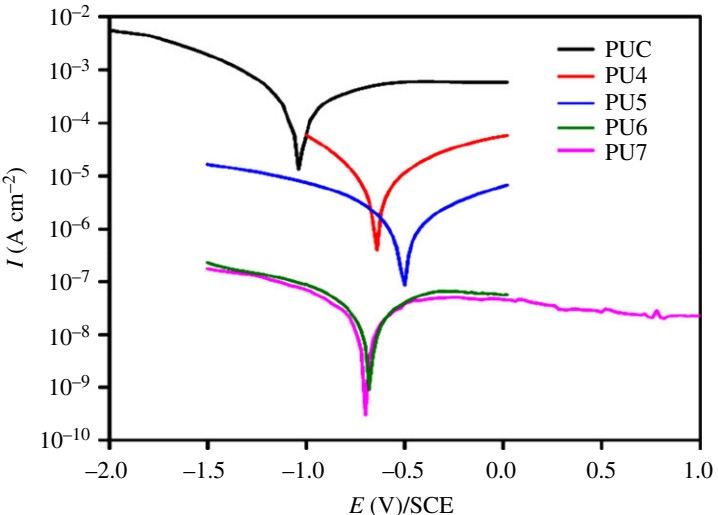

**Figure 5.** Tafel curves of various PU-coated steels after soaking for 30 days.

**Table 3.** Electrochemical parameters values of the various PU-coated steel after soaking for 30 days.

| coating | $I_{corr}$ (A cm$^{-2}$) | $\beta_c$ (V/dec) | $\beta_a$ (V/dec) | $R_p$ (k$\Omega$) | $E_{corr}$ (V) | CR (mm yr$^{-1}$) |
|---------|------------------|----------|----------|----------|----------|----------|
| PUC | $8.56 \times 10^{-6}$ | 0.71 | 0.105 | 1.21 | −1.04 | $9.93 \times 10^{-2}$ |
| PU4 | $4.92 \times 10^{-8}$ | 0.10 | 0.05 | 2.41 | −0.68 | $5.71 \times 10^{-4}$ |
| PU5 | $2.77 \times 10^{-9}$ | 0.011 | 0.011 | 2.78 | −0.68 | $9.02 \times 10^{-5}$ |
| PU6 | $7.55 \times 10^{-10}$ | 0.075 | 0.057 | 7.83 | −0.51 | $8.76 \times 10^{-6}$ |
| PU7 | $7.79 \times 10^{-10}$ | 0.55 | 0.72 | 7.85 | −0.51 | $9.05 \times 10^{-6}$ |

### 3.7.2. Potentiodynamic polarization

Potentiodynamic polarization measurement is a very effective technique for assessing the instant rate of corrosion of a material. Various PU coatings have been assessed by Tafel analysis to evaluate their corrosion rate (CR) after soaking for 30 days. Table 3 provides results of the polarization measurement showing the values of the electrochemical parameters. Generally, a greater $E_{corr}$ and lesser $I_{corr}$ and CR refer to better corrosion resistance [29]. Similarly, the polarization curves of the PU coatings are presented in figure 5. The corrosion potential ($E_{corr}$) of PU7 coated mild steel panels has shifted to more positive parts in comparison with PUC, PU4, PU5 and PU6. The $I_{corr}$ and $E_{corr}$ of the control (PUC) coating were $8.56 \times 10^{-6}$ A cm$^{-2}$ and −1.04 V, respectively. However, there was a considerable decrease in $I_{corr}$ to as low as $7.79 \times 10^{-10}$ A cm$^{-2}$ and an increase in $E_{corr}$ to −0.51 V for PU7. This revealed that the corrosion process was passive by PU7, which exhibited much higher corrosion protection ability in comparison with PUC.

The results of the polarization study revealed that the CR of PU7 is lowest in comparison with the other coatings as confirmed by its highest and lowest values for $E_{corr}$ and $I_{corr}$. These results also confirmed the influence of incorporating PET in the coatings for the prevention of the steel surface against corrosion. The corrosion current density was evaluated from the extrapolation of the Tafel slope by means of a GPES software in the Autolab PGSTAT 302N (Metrohm).

### 3.7.3. FTIR spectra of polyurethanes coatings

Referring to the FTIR spectra in figure 6, the O–H stretching band has disappeared and been replaced by N–H stretching and bending bands of urethane bond at 3348–3337 and 1508 cm$^{-1}$. The amide N–H in plane bending occurs at 1596 cm$^{-1}$. Band at 1715–1709 cm$^{-1}$ relates to hydrogen bonded –C=O stretching, in agreement to the formation of PU [31,32].

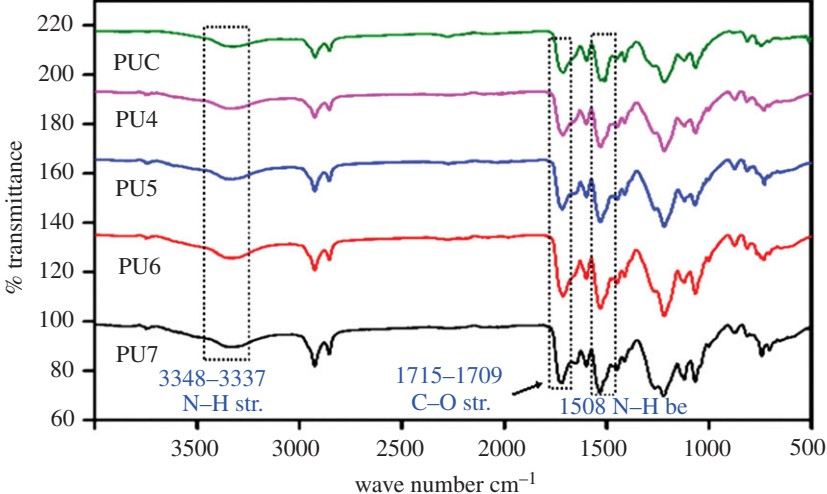

**Figure 6.** FTIR spectra of PUC, PU4, PU5, PU6, PU7 coatings.

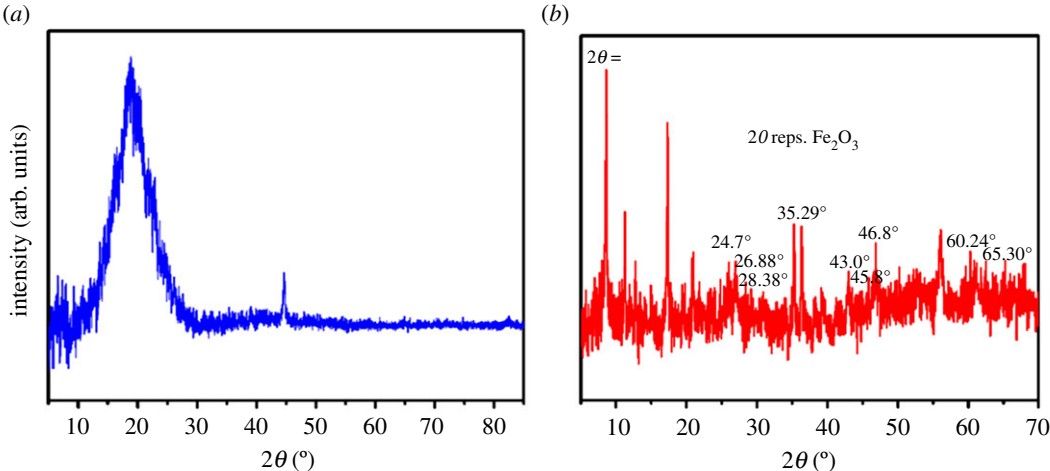

**Figure 7.** XRD pattern of (*a*) unexposed and (*b*) exposed PU-coated sample for 30 days in 3.5% NaCl solution.

### 3.7.4. XRD analysis of exposed and unexposed PU-coated sample

The PU-coated samples were analysed using XRD and the data is presented in figure 7. The pattern of unexposed sample is shown in (*a*), where the diffraction profiles show an amorphous broad shoulder, and a dispersed diffraction maximum at $2\theta = 20°$. This appears possibly that some soft–hard segment phase fraternization could occur in the system interrupting the crystallization of the soft segment. The presence of the amorphous and crystalline phases in the XRD diffractographs present in a real polymer, signified that polyurethane is a good polymer for industrial applications [33].

The data collection was recorded in the range of $2\theta = 5–89°$ with a step of 0.02°. In the respective curves, the *d*-spacing corresponding to the large peak(s) were calculated from Bragg's equation:

$$\lambda = 2(d - \text{spacing sin } \theta),$$

where *d* is the interplanar spacing, $2\theta$ is the X-ray scattering angle and $\lambda$ is the wavelength of the incident X-ray beam.

X-ray diffraction spectra were carried out on the substrate, for the identification of ferric oxide the JCPDS system was applied, and JCPDS file no. 85-0987 is noted after assessing the distance between the planes and their strength relative to each phase [34].

A common feature in the X-ray diffractogram obtained for the mild steel is the predominance of reflection lines equivalent to haematite ($Fe_2O_3$) [35]. The result of the XRD analysis of exposed PU sample represented by diffractograms figure 7*b* shows broad peaks at $2\theta$ angles around 24.74, 28.38, 35.29, 43.0, 45.8, 46.8, 56.8, 60.24 and 65.30°, which indicates certain degree of crystallinity with a

strong reflection lines at 4.069, 3.595, 3.142, 2.541, 2.101, 1.979, 1.939, 1.619, 1.535 and 1.427 Å, respectively, which agree with the results acquired in XRD study by Sharma & Jeevanandam et al. [34], Yunos et al. [36] Martínez et al. [37] and Trovati et al. [38]. These peaks are assigned to the scattering from PU chains with regular interplanar spacing [38]. From these results, it has been seen that the sample that was subjected to corrosion studies has shown some characteristic peaks that indicated that corrosion has taken place on the coated sample by the presence of $Fe_2O_3$ peaks on the sample. While on the other hand there was the absence of such peaks on the unexposed sample. This confirmed that corrosion has taking place after prolonged exposing the samples in 3.5% NaCl solution.

# 4. Conclusion

Polyols from palm olein and recycled PET were effectively synthesized and their properties characterized. Polyurethane coatings were prepared by reacting the polyols with methylene diphenyl diisocyanate (MDI) and characterized for thermal and anticorrosion properties. The formation of polyurethanes was confirmed by the formation of urethane linkages in the FTIR spectra. The PU coatings have exhibited significantly improved thermal stability and better anticorrosion properties with the incorporation of higher amount of PET into the palm olein alkyd.

Data accessibility. I have provided and uploaded my data in the Dryad database: https://doi.org/10.5061/dryad. 3ffbg79fz [39].

Authors' contributions. All the authors have contributed significantly in ensuring the success of the research work. A.A.A. carried out the practical research and the data analysis and compilation of the manuscript. Both the supervisors, S.N.G. and N.M.S., contributed in the designing and supervision of the research work and editing of the manuscript.

Competing interests. There is no conflict of interest to declare.

Funding. The project was funded by the University of Malaya.

Acknowledgements. We appreciate the University of Malaya for monetary funding under Impact-Orientated Interdisciplinary Research grant nos. IIRG006A-19IISS and PG311-2016A.

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
