## [Peer Review File · Royal Society Open Science]

Review History

RSOS-201087.R0 (Original submission)

Review form: Reviewer 1

Is the manuscript scientifically sound in its present form?

Yes

Are the interpretations and conclusions justified by the results?

Yes

Is the language acceptable?

Yes

Do you have any ethical concerns with this paper?

No

Have you any concerns about statistical analyses in this paper?

No

Recommendation?

Accept with minor revision (please list in comments)

Comments to the Author(s)

Dear Author,

This paper is suitable for publication but need some minor changes like in Experimental technique details such as Hydroxyl value, Acid value included in supplementary materials and remove from main article.

Review form: Reviewer 2

Is the manuscript scientifically sound in its present form?

No

Are the interpretations and conclusions justified by the results?

No

Is the language acceptable?

No

Do you have any ethical concerns with this paper?

No

Have you any concerns about statistical analyses in this paper?

No

Recommendation?

Major revision is needed (please make suggestions in comments)

Comments to the Author(s)

The author needs to give a detailed literature review for the study. There are 29 publications form Web of Science on the ulitisation of PET waste for polyurethane production. The authors need to give a strong justification on the needs, enhancement, significance of the present study as compared to the previous works.

There are many glaring grammatical errors, typographical, and mechanical errors throughout the manuscript. The structure of the manuscript needs to be rectified, where details of the methods are being discussed in the "Results and Discussion".

Specific comments:

Methodology:

The preparation of the polyurethane with different % of PET-based polyol is not revealed? The formulations are not presented? The reader is confused how the polyols C, 4,5,6,7 are being produced, what are the conditions and the reactant ratios? Which polyol is used for polyurethane production by reacting different percentages (what are the percentages?) of polyols with MDI? The methodology is vague.

Results and discussion:

Synthesis of palm olein/recycled-based polyols- There should be a discussion on the synthesis of the palm olein/recycled-based polyols. What are the reactions happened and how the polyols are being optimised? What are the reactions between the reactants, particularly the roles of the PET, Phthalic acid, ethylene glycol. Also, there should be a discussion on the reaction between the PET-based polyols with the isocyanate in the production of polyurethane.

The description in Results and Discussion from OHV, Acid Value to the first 7 sentences of the viscosity paragraph should be discussed in "Methodology" section.

Table 1: The acid value of the polyols are rather high, in which the PU industries are not favouring polyol products with high acid content, the acids react with isocyanate to form amide hard segments. Pls discuss

FTIR spectroscopy of polyols

The FTIR spectroscopy of the polyols should be made compared with the starting material. The highlighted functional groups like ester and OH band are present in the Monoglycerides, hence authors should make comparison to evident the reactions occur.

FTIR for polyols and PU-PIs indicate the important functional groups in the spectra

Table 3: how many replicates done on the PU coatings (n=?), what is the SD for the electrochemical parameters?

Page 4, right column, lines 15-21: urethane is linked contributing to a stronger intermolecular interaction.....-Should support by literature review.

Decision letter (RSOS-201087.R0)

Dear Dr Adamu:

Title: Thermal and anticorrosive properties of PU coatings derived from recycled PET and palm olein based polyols

Manuscript ID: RSOS-201087

The editor assigned to your manuscript has now received comments from reviewers. We would like you to revise your paper in accordance with the referee and Subject Editor suggestions which can be found below (not including confidential reports to the Editor). Please note this decision does not guarantee eventual acceptance.

Please submit your revised paper before 16-Sep-2020. Please note that the revision deadline will expire at 00.00am on this date. If we do not hear from you within this time then it will be assumed that the paper has been withdrawn. In exceptional circumstances, extensions may be possible if agreed with the Editorial Office in advance. We do not allow multiple rounds of revision so we urge you to make every effort to fully address all of the comments at this stage. If deemed necessary by the Editors, your manuscript will be sent back to one or more of the original reviewers for assessment. If the original reviewers are not available we may invite new reviewers.

On behalf of the Subject Editor Professor Anthony Stace and the Associate Editor Dr Chaohua Cui.

RSC Associate Editor:
Comments to the Author:
(There are no comments.)

RSC Subject Editor:
Comments to the Author:
(There are no comments.)

Reviewers' Comments to Author:

Reviewer: 1

Comments to the Author(s)

Dear Author,

This paper is suitable for publication but need some minor changes like in Experimental technique details such as Hydroxyl value, Acid value included in supplementary materials and remove from main article.

Reviewer: 2

Comments to the Author(s)

The author needs to give a detailed literature review for the study. There are 29 publications from Web of Science on the utilisation of PET waste for polyurethane production. The authors need to give a strong justification on the needs, enhancement, significance of the present study as compared to the previous works.

There are many glaring grammatical errors, typographical, and mechanical errors throughout the manuscript. The structure of the manuscript needs to be rectified, where details of the methods are being discussed in the "Results and Discussion".

Specific comments:

Methodology:

The preparation of the polyurethane with different % of PET-based polyol is not revealed? The formulations are not presented? The reader is confused how the polyols C, 4,5,6,7 are being produced, what are the conditions and the reactant ratios? Which polyol is used for polyurethane production by reacting different percentages (what are the percentages?) of polyols with MDI? The methodology is vague.

Results and discussion:

Synthesis of palm olein/recycled-based polyols- There should be a discussion on the synthesis of the palm olein/recycled-based polyols. What are the reactions happened and how the polyols are being optimised? What are the reactions between the reactants, particularly the roles of the PET, Phthalic acid, ethylene glycol. Also, there should be a discussion on the reaction between the PET-based polyols with the isocyanate in the production of polyurethane.

The description in Results and Discussion from OHV, Acid Value to the first 7 sentences of the viscosity paragraph should be discussed in "Methodology" section.

Table 1: The acid value of the polyols are rather high, in which the PU industries are not favouring polyol products with high acid content, the acids react with isocyanate to form amide hard segments. Pls discuss

FTIR spectroscopy of polyols

The FTIR spectroscopy of the polyols should be made compared with the starting material. The highlighted functional groups like ester and OH band are present in the Monoglycerides, hence authors should make comparison to evident the reactions occur.

FTIR for polyols and PU-PIs indicate the important functional groups in the spectra

Table 3: how many replicates done on the PU coatings (n=?), what is the SD for the electrochemical parameters?

Page 4, right column, lines 15-21: urethane is linked contributing to a stronger intermolecular interaction.....-Should support by literature review.

Author's Response to Decision Letter for (RSOS-201087.R0)

See Appendix A.

RSOS-201087.R1 (Revision)

Review form: Reviewer 2

Is the manuscript scientifically sound in its present form?

Yes

Are the interpretations and conclusions justified by the results?

Yes

Is the language acceptable?

Yes

Do you have any ethical concerns with this paper?

No

Have you any concerns about statistical analyses in this paper?

No

Recommendation?

Accept with minor revision (please list in comments)

Comments to the Author(s)

The revised manuscript has shown much improvement, it can be accepted for publication with the following revision made prior to publication.

Typo error: Abstract: page 1, line 25: by XRD analysis,--comma should be full-stop

Page 1, line 25: up to 15%---15%w/w?

Introduction: page 1, right column, line 46: they are mostly produced from polyether polyols---should be "they are mostly produced from polyether and polyester polyols"

Introduction: page 1, right column, line 50: polyurethans are also widely used as coatings.16,---remove comma after reference 16

Page 2, introduction: et al. should be italic, please check, there are at least 7 typo errors on "et al."

Page 2: Introduction last paragraph: is first synthesised---should be was synthesised

The new polyol is formulated---should be was formulated

Suggest to remove the last sentence of the introduction "These justify the work as novel and significant".

The last sentence should be "We are the first to report the direct incorporation of PET into a palm oil-based alkyd to form polyol for making new polyurethane coatings.---add "coatings".

Results and Discussion: The reader would want to learn more on the synthesis of the PET polyol using recycled PET. Suggest to provide reaction schemes to illustrate the reactions on how palm olein reacted with glycerol, phthalic anhydride, ethylene glycol and followed by PET.

Also describe on what basis the specific amounts of PET are selected (85.6, 106, 128, 132g).

Conclusion: The PU coatings exhibited excellent physico-chemical and outstanding corrosive properties---suggest to revise to "The PU coatings exhibited improved thermal stability and corrosive properties" This is due to the physico-chemical properties measured are only FTIR and nothing to evidence its improved physico-chemical properties. But instead TGA was analysed.

Also suggest not to use "superior" since the comparison only made on the control coating samples prepared and not a standard material/market product.

Decision letter (RSOS-201087.R1)

Dear Professor Gan:

Title: Thermal and anticorrosive properties of PU coatings derived from recycled PET and palm olein based polyols

Manuscript ID: RSOS-201087.R1

Thank you for submitting the above manuscript to Royal Society Open Science. On behalf of the Editors and the Royal Society of Chemistry, I am pleased to inform you that your manuscript will be accepted for publication in Royal Society Open Science subject to minor revision in accordance with the referee suggestions. Please find the reviewers' comments at the end of this email.

The reviewers and handling editors have recommended publication, but also suggest some minor revisions to your manuscript. Therefore, I invite you to respond to the comments and revise your manuscript.

Because the schedule for publication is very tight, it is a condition of publication that you submit the revised version of your manuscript before 05-Nov-2020. Please note that the revision deadline will expire at 00.00am on this date. If you do not think you will be able to meet this date please let me know immediately.

Kind regards,
Dr Laura Smith
Publishing Editor, Journals

On behalf of the Subject Editor Professor Anthony Stace and the Associate Editor Dr Chaohua Cui.

RSC Associate Editor:
Comments to the Author:
(There are no comments.)

RSC Subject Editor:
Comments to the Author:
(There are no comments.)

Reviewer comments to Author:
Reviewer: 2

Comments to the Author(s)

The revised manuscript has shown much improvement, it can be accepted for publication with the following revision made prior to publication.

Typo error: Abstract: page 1, line 25: by XRD analysis,--comma should be full-stop

Page 1, line 25: up to 15%---15%w/w?

Introduction: page 1, right column, line 46: they are mostly produced from polyether polyols---should be "they are mostly produced from polyether and polyester polyols"

Introduction: page 1, right column, line 50: polyurethans are also widely used as coatings.16,---remove comma after reference 16

Page 2, introduction: et al. should be italic, please check, there are at least 7 typo errors on "et al."

Page 2: Introduction last paragraph: is first synthesised---should be was synthesised

The new polyol is formulated---should be was formulated

Suggest to remove the last sentence of the introduction "These justify the work as novel and significant".

The last sentence should be "We are the first to report the direct incorporation of PET into a palm oil-based alkyd to form polyol for making new polyurethane coatings.---add "coatings".

Results and Discussion: The reader would want to learn more on the synthesis of the PET polyol using recycled PET. Suggest to provide reaction schemes to illustrate the reactions on how palm olein reacted with glycerol, phthalic anhydride, ethylene glycol and followed by PET.

Also describe on what basis the specific amounts of PET are selected (85.6, 106, 128, 132g).
Conclusion: The PU coatings exhibited excellent physico-chemical and outstanding corrosive properties---suggest to revise to “The PU coatings exhibited improved thermal stability and corrosive properties” This is due to the physico-chemical properties measured are only FTIR and nothing to evidence its improved physico-chemical properties. But instead TGA was analysed.
Also suggest not to use “superior” since the comparison only made on the control coating samples prepared and not a standard material/market product.

Author's Response to Decision Letter for (RSOS-201087.R1)

See Appendix B.

Decision letter (RSOS-201087.R2)

Dear Professor Gan:

Title: Thermal and anticorrosion properties of PU coatings derived from recycled PET and palm olein-based polyols
Manuscript ID: RSOS-201087.R2

It is a pleasure to accept your manuscript in its current form for publication in Royal Society Open Science. The chemistry content of Royal Society Open Science is published in collaboration with the Royal Society of Chemistry.

On behalf of the Subject Editor Professor Anthony Stace and the Associate Editor Professor Chaohua Cui.

RSC Associate Editor
Comments to the Author:
(There are no comments.)

Reviewer(s)' Comments to Author:

Appendix A

Reviewers' comments

Comments	Response
Reviewer 1	
This paper is suitable for publication but need some minor changes like in Experimental technique details such as Hydroxyl value, Acid value included in supplementary materials and remove from main article.	Noted. The hydroxyl value and acid value have been determined by standard test methods, as briefly described under the Experimental section.
Reviewer 2	
The author needs to give a detailed literature review for the study. There are 29 publications from Web of Science on the utilization of PET waste for polyurethane production. The authors need to give a strong justification on the needs, enhancement, significance of the present study as compared to the previous works. There are many glaring grammatical errors, typographical, and mechanical errors throughout the manuscript. The structure of the manuscript needs to be rectified, where details of the methods are being discussed in the "Results and Discussion".	The Introduction section has been revised and previous published works on utilizing PET waste for polyurethane were discussed. In the present work, an alkyd is first synthesised from palm olein, glycerol and phthalic anhydride. Waste PET from soft drink bottles and ethylene glycol were introduced into the alkyd at high temperature. Simultaneous depolymerisation of PET by ethylene glycol and transesterification with alkyd occur to form a viscous product (palm olein/PET polyols). The new polyol is formulated to produce anticorrosion PU coating. We are the first to report the direct incorporation of PET into a palm oil-based alkyd to form polyol for making new PU coatings. These justify the work as novel and significant. Grammatical/typing errors have been corrected.
Specific comments	
Methodology:	
The preparation of the polyurethane with different % of PET-based polyol is not revealed? The formulations are not presented? The reader is confused how the polyols C, 4,5,6,7 are being produced, what are the conditions and the reactant ratios? Which polyol is used for polyurethane production by reacting different percentages (what are the percentages?) of polyols with MDI? The methodology is vague.	Details of the polyols preparation and PU formulations are provided. Amounts of reactants for the synthesis are given and the weights of PET in the four polyols are stated. Typically, 11 g of the polyol is dissolved in 27 g solvent, and mixed with 16 g MDI and 0.021 g tin catalyst to form a total of 59 g of coating solution.
Results and discussion:	
Synthesis of palm olein/recycled PET-based polyols- There should be a discussion on the synthesis of the palm olein/recycled PET-based polyols. What are the reactions happened and how the polyols are being optimised? What are the reactions between the reactants, particularly the roles of the PET, Phthalic acid, ethylene glycol. Also, there should be a discussion	An alkyd is first synthesised from palm olein, glycerol and phthalic anhydride. PET and ethylene glycol were added into the alkyd at high temperature, depolymerisation of PET by ethylene glycol and transesterification occur simultaneously. The polyurethane is formed from the condensation reactions of the hydroxyl groups of the polyols with the isocyanate groups of MDI.

on the reaction between the PET-based polyols with the isocyanate in the production of polyurethane.	
The description in Results and Discussion from OHV, Acid Value to the first 7 sentences of the viscosity paragraph should be discussed in "Methodology" section.	The OHV, Acid Value and the viscosity are discussed under Experimental section.
Table 1: The acid values of the polyols are rather high, in which the PU industries are not favouring polyol products with high acid content, the acids react with isocyanate to form amide hard segments. Pls discuss	These alkyd polyols have AV of 15.5 ± 0.5 mg KOH g ⁻¹ , which is <3% OHV (535 ± 10). Residual -COOH groups in the alkyd could react with amino group of polyurethane chain to produce amide, which could contribute to film hardness.
FTIR spectroscopy of polyols The FTIR spectroscopy of the polyols should be made compared with the starting material. The highlighted functional groups like ester and OH band are present in the Monoglycerides, hence authors should make comparison to evident the reactions occur.	FTIR spectra of the polyols are shown in Figure 1. The -OH broad band is seen at 3450 cm^{-1} . A band close to 1735 cm^{-1} is accredited to C=O group of ester. Bands near 1656 cm^{-1} corresponds to aromatic ring. The -CH group bending vibrations present at frequencies near 1376 cm^{-1} and 1458 cm^{-1} . The presence of a strong band at 1072 cm^{-1} indicated that PET was incorporated into the polyol. However, the key bands of hydroxyl and carbonyl stretching are seen as broad peaks at 3450 cm^{-1} and 1735 cm^{-1} respectively, confirming the formation of polyester polyol. ^{15,21} .
FTIR for polyols and PU-PIs indicate the important functional groups in the spectra Table 3: how many replicates done on the PU coatings (n=?), what is the SD for the electrochemical parameters?	All the polyols have similar FTIR spectra where -OH and ester are the important functional groups. The positions of those peaks were discussed. The functional groups of PU samples are indicated in the spectra. We have not carried out error analysis.
Page 4, right column, lines 15-21: urethane is linked contributing to a stronger intermolecular interaction.... -Should support by literature review.	Agree to remove the speculative statement.

Appendix B

Upload a file "Response to Referees" in "Section 6 - File Upload".

Reviewer comments to Author:

Reviewer: 2

We thank the reviewer for pointing out the errors for us to improve the manuscript.

Comments to the Author(s)

The revised manuscript has shown much improvement, it can be accepted for publication with the following revision made prior to publication.

Typo error: Abstract: page 1, line 25: by XRD analysis,--comma should be full-stop

Page 1, line 25: up to 15%---15%w/w?

15% w/w

Introduction: page 1, right column, line 46: they are mostly produced from polyether polyols--
-should be "they are mostly produced from polyether and polyester polyols"

Introduction: page 1, right column, line 50: polyurethans are also widely used as
coatings.¹⁶,---remove comma after reference 16

Page 2, introduction: et al. should be italic, please check, there are at least 7 typo errors on
"et al."

Corrections done as suggested:

They are mostly produced from polyether and polyester polyols derived from petrochemicals.¹⁵

... are also widely used as coatings.¹⁶ Corrosion....

Corrected to "*et al.*"

Page 2: Introduction last paragraph: is first synthesised---should be was synthesised

The new polyol is formulated---should be **was formulated**

Suggest to remove the last sentence of the introduction "~~These justify the work as novel and significant~~". Deleted as suggested.

The last sentence should be "We are the first to report the direct incorporation of PET into a palm oil-based alkyd to form polyol for making new **polyurethane coatings**.---add "coatings".

Corrections done as suggested and highlighted in yellow.

Results and Discussion: The reader would want to learn more on the synthesis of the PET polyol using recycled PET. Suggest to provide reaction schemes to illustrate the reactions on how palm olein reacted with glycerol, phthalic anhydride, ethylene glycol and followed by PET.

A scheme is included to show the plausible reactions.

Also describe on what basis the specific amounts of PET are selected (85.6, 106, 128, 132g).

Incorporation of PET into the alkyd has led to increase in the viscosity of the final polyol, and the polyol would become non flowable if more than 132 g PET was used.

Conclusion: The PU coatings exhibited excellent physico-chemical and outstanding corrosive properties---suggest to revise to “The PU coatings exhibited improved thermal stability and corrosive properties” This is due to the physico-chemical properties measured are only FTIR and nothing to evidence its improved physico-chemical properties. But instead TGA was analysed. Also suggest not to use “superior” since the comparison only made on the control coating samples prepared and not a standard material/market product.

The conclusion has been rewritten:

Polyols from palm olein alkyd and recycled PET were effectively synthesized and their properties characterized. Anticorrosion polyurethane coatings were prepared by reacting the polyols with methylene diphenyl diisocyanate (MDI) and characterized for anticorrosion properties. The formation of polyurethanes was confirmed by the FTIR spectra of the prepared polyurethane coatings. The PU coatings have exhibited improved thermal stability and better anti corrosion properties with the incorporation of higher amount of PET into the palm olein alkyd.